# Portraying Animal Cruelty: A Thematic Analysis of Australian News Media Reports on Penalties for Animal Cruelty

**DOI:** 10.3390/ani12212918

**Published:** 2022-10-25

**Authors:** Rochelle Morton, Michelle L. Hebart, Rachel A. Ankeny, Alexandra L. Whittaker

**Affiliations:** 1School of Animal and Veterinary Sciences, The University of Adelaide, Roseworthy, SA 5371, Australia; 2School of Humanities, The University of Adelaide, Adelaide, SA 5005, Australia

**Keywords:** animal welfare, animal cruelty, news media, penalties, animal law, Facebook

## Abstract

**Simple Summary:**

News media is one of the major sources of publicly available information on animal welfare law enforcement. It has previously been established that the media are strong influences of public perceptions. Therefore, it is possible that news reports on animal cruelty offences are shaping public understanding of penalties for animal cruelty. To understand how penalties are portrayed in the media, we collected 71 Australian news articles which reported on penalties for animal cruelty over a 6-month period from 1 June 2019 to 1 December 2019. Using thematic analysis, three themes were identified: (1) laws are not good enough; (2) laws are improving; and (3) reforms are unnecessary. A connection between public perceptions, media reporting and statutory reform efforts to increase maximum penalties is proposed, which potentially could explain why the Australian public appear displeased with the penalties handed down by courts for animal cruelty offences. Further sociological research is required to confirm this theory.

**Abstract:**

Media portrayals of animal cruelty can shape public understanding and perception of animal welfare law. Given that animal welfare law in Australia is guided partially by ‘community expectations’, the media might indirectly be influencing recent reform efforts to amend maximum penalties in Australia, through guiding and shaping public opinion. This paper reports on Australian news articles which refer to penalties for animal cruelty published between 1 June 2019 and 1 December 2019. Using the electronic database Newsbank, a total of 71 news articles were included for thematic analysis. Three contrasting themes were identified: (1) laws are not good enough; (2) laws are improving; and (3) reforms are unnecessary. We propose a penalty reform cycle to represent the relationship between themes one and two, and ‘community expectations’. The cycle is as follows: media reports on recent amendments imply that ‘laws are improving’ (theme two). Due to a range of inherent factors in the criminal justice system, harsher sentences are not handed down by the courts, resulting in media report of ‘lenient sentencing’ (theme one). Hence, the public become displeased with the penal system, forming the ‘community expectations’, which then fuel future reform efforts. Thus, the cycle continues.

## 1. Introduction

Animal welfare law seeks to regulate human conduct towards animals by codifying what society deems as unacceptable treatment of animals. This jurisprudential model recognizes that animals can experience pain and suffering, and that their interests in avoiding these experiences are considered morally relevant by society [1]. Protection of animals is provided in animal welfare legislation through the provision of offences for cruelty and the establishment of a duty of care. Given that animal welfare is a societally important issue, it is generally accepted that governments will legislate animal welfare in the public interest [2,3], meaning that community expectations can influence the direction and scope of animal welfare legislation. This is evident from recent reform efforts to state-based Animal Welfare Acts in Australia, where several of these amendments make reference to aligning the legislation with the ‘community expectations’ [4,5]. However, as Geysen et al. [4] noted, the nature of such ‘expectations’ are ultimately unknown, and thus their incorporation in legislative reform may involve no more than paying them ‘lip service’ or surmising their nature.

Given that a major source of public information on crime is news media sources [6], this outlet is a logical starting point in developing an understanding of community expectations of animal cruelty as a crime. As stated by a New South Wales Magistrate regarding sentencing for animal cruelty matters “public confidence in the system of justice administered by the courts is vital to society and is enhanced by informed public debate. This requires responsible media” [7]. News media is the central repository of information that the public use to build their definition of a crime [8]. For example, it has been established that news media can shape public understanding and perceptions of crimes [6], influence public knowledge and attitude towards issues [9], and shape policy implementation and public debate [10,11,12,13,14]. News media inform, whilst providing a platform for discussion, yet also selectively choose issues to report on based on public interest, and have the power to shape public perceptions and attitudes through the way information is presented [15]. Thus, the news media has a substantial role in setting the public policy scene, as it can make some issues salient, whilst marginalizing others [15,16]. This process, of highlighting issues and excluding others to promote a certain mode of thought, is known as agenda-setting [17].

The theory of agenda-setting has two levels. The first level focuses on the level of coverage an issue receives, as issues emphasized by the media will translate into issues the public believe are important [17]. Whereas second level agenda setting focuses not on the prevalence of information, but how that information is discussed in an effort to understand how the public may perceive the issues which have captured their attention [17]. The second level of agenda-setting will be the focus in this paper. It should be noted that agenda-setting is not the result of journalists trying to control public perceptions and attitudes, instead it is caused by the necessity to capture focus, as public attention can only be directed to a few issues in a short period of time [18]. In addition, agenda-setting is most relevant when readers have little direct experience with that issue [19,20]. The majority of the public do not have any direct or indirect exposure to crimes of animal cruelty [21]. Therefore, based on agenda-setting theory, community expectations about the nature of animal cruelty, and its handling by the criminal justice system, could be guided by media portrayals.

In an analysis of media reporting of animal hoarding cases in the United States, Arluke et al. [22] established that whilst factual information was presented in articles, it was often distorted by presenting articles in a way that would generate more public interest. Additionally, there was a tendency to describe severe cases of animal hoarding, which are the minority, making cases of animal hoarding appear substantially worse than they actually are [22]. As stated by Grugan [21] from their analysis of companion animal cruelty reporting, “the media decides which forms of cruelty are worthy of condemnation, which animals are worthy of sympathy, which offences are worthy of dramatic and emotive descriptions, and when it is appropriate to advocate on behalf of animals and when it is not”. However, both studies were US-focused with extrapolation to the Australian context being problematic given the domestic nature of animal welfare legislation. Furthermore, a factor that was not extensively analyzed in Arluke et al. [22] and Grugan [21] research on animal cruelty reports was the penalties for offences, in the form of custodial sentences and monetary fines. Maximum penalties, as set by parliaments, are found in Animal Welfare Acts and provide the sentencing courts a benchmark against which the gravity of an offence should be measured when handing down penalties to offenders [23]. Increases to maximum penalties have been a regular focus for reform to state-based animal welfare legislation in Australia, in an attempt to align with ‘community expectations’ [4,5]. This occurred in Queensland in 2001 [24], South Australia in 2008 [25], Victoria in 2012 [26], Australian Capital Territory in 2019 [27] and Northern Territory introducing their proposal in 2020 [28]. In the federated Australian system, each state and territory has set different maximum penalties for both duty of care breaches and deliberate cruelty. Considering the more serious (aggravated) offences only the maximum penalty ranges from two years in prison to five years (see Morton et al. [29] for a detailed account of all custodial and monetary maximums for each offence across the Australian states and territories).

The process of setting maximum penalties and handing down penalties to offenders is carried out separately; parliamentarians set the maximum penalties laid out in Acts of parliament, and the court system, through judicial officers, decides the specific penalty for an offence. Judges determine penalties based on a balanced consideration of the details of the crime, the penalties laid out in the Act, as well as individual defendant-related factors. In theory, the two processes should work symbiotically, where the parliamentary intention to ‘get tough’ on offenders through increasing the maximum penalties [30,31] is translated into harsher sentences imposed by the sentencing courts [4,23,32]. However, despite two decades worth of reforms resulting in maximum penalty increases in Australia, reports on public sentiment around sentencing of animal cruelty suggests that they are still not harsh enough [33,34]. This could imply that parliamentary intent is failing to translate to the court system. There is some limited evidence for this with the finding that only 10% of the maximum penalties for animal cruelty are being used in South Australian courts [5]. However, an alternative explanation is that ‘community expectations’ are not aligning with the intent of the legislation, in that the harsher penalties the public are in favor of [33,34], are not being made use of by the justice system in the opinion of the public. In this context, the term ‘community expectations’ is being used to describe collective public opinion, whether perceived or gleaned from sociological research. Reasons for this disconnect could be that community expectations towards animal law enforcement are increasing at a rate the legislation cannot, or that the community expect too much from the animal law legal system given resource and funding availability [35,36].

Given the lack of ongoing access to information on public opinion around animal cruelty penalties it is impossible to assess their nature, and how they are reflected in the legislation. However, based on second level agenda-setting theory, it is likely that analyzing the themes present in animal cruelty media reports can give an indication of these ‘community expectations’, as the themes present are likely the building blocks of public opinion given the influential nature of the media [6,9,10,11,12,13,14]. Therefore, this study builds upon Arluke et al. [22] and Grugan [21] research and investigates the emerging themes from Australian media reports of penalties for animal welfare offences. Using thematic analyses, the predominant themes present in animal cruelty media reports, that refer to the penalties in some capacity, are identified and discussed in terms of being indicators of public opinion. Since agenda-setting analyses have found to be consistently correlated with the public agenda [37], this study provides a viewpoint on how the media may be influencing ‘community expectations’, and in turn animal welfare legislation in Australia.

## 2. Materials and Methods

### 2.1. Newspaper Selection

This study focused on Australian news reporting nationally despite the state-based approach to animal welfare legislation, as previously it has been identified that all the state and territory-based Animal Welfare Acts in Australia are similar in terms of penalty structure and sentencing principles [29]. Therefore, articles were selected from all online national Australian print media sources. Print media remains a widely accessible media source in Australia that plays a significant role in influencing issues in society [38]. Print news media is important for political decision making by downplaying some issues and diverting that attention to others [39]. Therefore, with print media becoming widely available to a diverse demographic of Australians through electronic sources and social media sharing [40]; newspaper articles remain a popular and widely available source of current affairs relating to animal cruelty offences in Australia. All newspapers from all Australian states and territories were selected, excluding any magazines, newsletters, journals, or blog posts from the search. A total of 533 Australian newspapers available on the electronic database Newsbank [41] were included in the search. Metropolitan and rural newspapers were included to reduce any demographic biases between the two news sources and identify a newspaper sample with broad readership demographics and political orientations.

### 2.2. Search Strategy

Similar to Grugan [21] content analysis methodology, a 6-month period from 1 June 2019 to 1 December 2019 was selected. This period was prior to the emerging COVID-19 news articles in the Australia media. Articles were retrieved from Newsbank [41] using the following search criteria: “animal* (lead/first paragraph)” AND “welfare OR abuse OR cruel* OR violent* (all text)” AND “penalty* OR case OR offence OR prosecution OR crime* OR sentence OR case OR charge OR law OR illegal OR legal OR justice OR prison OR jail OR fine (all text)” OR “animal cruelty case” OR “welfare offence” OR “cruelty offence” OR “cruelty charges”.

### 2.3. Eligibility Criteria

Articles accepted for analysis were those that included specific discussions on sentencing in animal cruelty cases, or articles which discussed penalties in general terms. The latter included reports on animal welfare legislation reform or making referral to maximum penalties for animal welfare offences. The nature of the articles included varied from court reports to those discussing penalties arising under the eight state and territory-based animal protection acts in Australia. Articles discussing penalties arising for offences towards animals under other legislation, such as crimes acts, domestic animal acts, or wildlife acts, were excluded due to their differing enforcement models and penalty types.

Articles were excluded if they did not make any reference to penalties or animal welfare offences. Furthermore, any ‘letters of editors’ or public submissions were excluded as they often included multiple topics and lacked a focus on animal cruelty cases. The ‘year in review’ articles were excluded, as these revisited already published articles. Finally, all duplicates were excluded. The search identified 787 articles; 128 remained after articles were screened for the eligibility criteria. A further 57 articles were excluded after examining for duplicates. Thus, a total of 71 articles remained for analysis.

### 2.4. Analysis

A thematic analysis was performed following the methods of Braun and Clarke [42]. Articles were imported into Nvivo 12 [43] for coding. Both the semantic and latent attributes of the data were considered, and an inductive coding approach was taken, where codes and themes were developed from the data content. The version of thematic analysis used in this study, reflexive thematic analysis, advocates against the use of measures of inter-rater reliability and other such coding practices as a measure of quality [44]. Therefore, coding was performed by one coder (RM) in the absence of a codebook. Following initial coding, the codes were returned to, and revised as the coding process proceeded. Codes were then clustered into candidate themes to give some indication of their prevalence, and test their value in giving an overall account of the data [45]. Six themes were initially identified, however after thematic maps identified theoretical overlap between the themes, they were collapsed into three final themes.

## 3. Results

Three salient themes were generated from the analysis: (1) current animal welfare laws are not good enough and need reform; (2) laws are improving, and animal welfare is being taken seriously; (3) and reforms are too harsh and unnecessary. The majority of the data were included in these three predominant themes.

### 3.1. Theme 1: ‘Animal Welfare Laws Are Failing and Need Reform’: Action Is Needed

The first theme captured the need to take action to combat the failings of animal welfare laws. These ‘failures’ were often ascribed to legislation not being tough enough to adequately protect animals, and more often, expressed the need for legislative reform. Many articles made statements similar to the following:

“*For many years, I, too, have questioned the way animals are treated and just why animal welfare laws do not go far enough*”(stated by journalist, Queensland Times, 14 August 2019)

Overall, this theme was the most common in the dataset, expressed by approximately half the included articles. In addition to these articles discussing the need for legislative reform, similarly to Parliamentarians, the authors would also refer to community expectations:

“*She also called on the courts to enforce sentences more in line “with community expectation” in animal cruelty cases*”(quoted from RSPCA ACT spokesperson, Canberra Times, 20 September 2019)

“*Our pets are much-loved members of our families. The community has no tolerance for this kind of offending*”(quoted from Sentencing Advisory Council spokesperson, Central Queensland News, 30 August 2019)

In making such statements, authors infer that the process of justice is failing due to disparity between community expectations and the sentencing reality. Animal welfare is generally accepted as an issue in which governments legislate in the public interest [2,3]. Thus, alignment of community expectations with the legislative objective should occur. The majority of references to this disparity followed with comments regarding the leniency of penalties for animal welfare offences.

“*Animal bans are not enough. Many members of the community would like to see jail terms imposed for severe animal cruelty cases, and more cases also need to be brought to court*”(quoted from pet rescue spokesperson, Central Queensland News, 30 August 2019)

“*The attack has prompted calls for harsher penalties for acts of animal cruelty, including mandatory jail time*”(stated by journalist, The Examiner (Launceston), 28 July 2019)

These statements imply that to correct the disparity, the penalties for animal welfare offences need to be harsher. Maximum penalty increases are not an uncommon reform in Australia, however it has been reported that penalty increases are more of a symbolic gesture to reflect this notion of “getting tough” rather than effecting meaningful changes by the sentencing courts [5]. Therefore, unsurprisingly, many articles also referred to lenient sentences handed out in court.

“*We have a terrible history in Tasmania of absolutely pathetic punishments for people doing barbaric things to animals*”(quoted from wildlife sanctuary spokesperson, The Mercury (Hobart), 27 July 2019)

“*The sentence handed down last week certainly does not reflect the seriousness of this crime… It’s time that we started to impose these maximum sentences and treat cases of cruelty to animals as the serious crimes that they are*”(stated by journalist, Northern Miner (Townsville), 11 July 2019)

“*This act of cruelty is sickening, we don’t want this to become another case of lenient sentencing that will do nothing to deter further acts of extreme animal mistreatment*”(quoted from Member of Parliament, Midstate Observer (Orange), 18 June 2019)

“*SERIOUSLY, what does someone have to do to an animal to land themselves a bed in jail?*”(quoted from RSPCA QLD spokesperson, The Sunday Mail (Queensland), 7 August 2019)

Pairing the two sentiments of penalty increases within statute and harsher imposition of sentences at the court level, the messaging appeared to be that increasing the statutory maximums should result in harsher sentence imposition by the courts, with ‘harsher’ generally considered as terms of imprisonment, with statements like the following:

“*When we think about tougher penalties for cruelty, we think about jail*”(quoted from RSPCA QLD spokesperson, The Sunday Mail (Queensland), 7 August 2019)

In addition, it was also implied that action was needed to strengthen animal welfare legislation since acts of cruelty were getting worse and becoming more common.

“*Animal cruelty is getting worse in this State, and it seems to be a very bad year this year and I hope this is a wakeup call*”(quoted from farm sanctuary spokesperson, The Advocate (Tasmania), 27 July 2019)

“*RSPCA inspectors are rescuing almost two animals a day on average as the number of reports of neglect increased by more than 200 since last year*”(stated by journalist, Sunday Mail (Adelaide), 4 August 2019)

Extracts such as these imply that without further action animal cruelty will continue to become a larger and more severe problem in society. They speak to a need to act before it’s too late, creating a sense of urgency. Similarly, the link between human and animal violence, as established in the literature [46,47,48,49,50,51,52,53], was addressed on numerous occasions with the view propounded that animal cruelty should be stopped before it progresses into human violence.

“*While this incident is shocking enough in itself, it overlooks the link between animal abuse and domestic violence: a causal link that has been researched extensively but is often ignored when animal abuse incidents are treated in isolation from their wider social implications… Alarmingly, there is also the documented fact that children who are exposed to animal abuse absorb and normalize this behavior, thus perpetuating a cycle of violence*”(stated by journalist, Magnet Eden (New South Wales), 27 June 2019)

“*The research, not just in Australia, has shown that deliberate, premeditated animal cruelty, which this was, is a precursor to other types of violence… So, we really need to take this seriously*”(quoted from RSPCA QLD spokesperson, Daily Mercury (Queensland), 24 June 2019)

Pairing this urgency to act before it’s too late with the threat of future human violence, could target a different demographic than the usual ‘animal-lovers’, potentially resulting in visceral reactions from a more diverse, larger group of readers.

Finally, a relatively small number of articles referred to the idea that animals are receiving no justice from the criminal justice system. They were often described as “defenseless” and “voiceless” creatures, which are inadequately represented and protected during the court process. This idea was often related to the ‘lenient sentences’ addressed earlier, since the court failure to impose a harsh sentence on the offender, fails to afford justice to the animal.

“*Unfortunately, these defenseless animals had no voice and received no justice*”(quoted from RSPCA ACT spokesperson, The Canberra Times, 20 September 2019)

### 3.2. Theme 2: ‘We Take Animal Welfare Seriously and Are Reforming Laws’: Action Is Being Taken

The second, contrasting, theme contained a number of extracts suggesting animal welfare legislation is improving, and that animal welfare is a clear priority for the Australian state and territory governments. In comparison with the first theme, this theme represented more positive messages, being ‘we hear your concerns and are taking action’. This theme was most prevalent in the dataset when reporting penalty increases, whether to the maximum penalties through amendments (e.g., “Legislation introduces tough new penalties”—stated by journalist, The Canberra Times, 27 September 2019) or when securing a high sentence in court (e.g., “It’s the longest jail term imposed for an RSPCA NSW prosecution”—stated by journalist, Liverpool Champion New South Wales, 2 July 2019).

Accounts of amendments in this theme were primarily focused on the maximum penalties. The penalties were either quoted directly with the monetary value of a fine and the term of a custodial sentence, or they were summarized as “tough new penalties”.

“*Dog owners who leave their pet in a hot car even for a few minutes will face whopping fines of $40,000 and a year’s jail under beefed-up laws targeting animal cruelty—and stupidity*”(stated by journalist, The Sunday Mail Queensland, 16 September 2019)

Commonly, such articles would focus on amendments in general, and not discuss the penalties being applied in court. These articles would link the ‘community expectations’ to these amendments, implying that authorities are listening to the community and attempting to close the gap between public expectations and outcomes.

There was some discussion around penalties in action, in terms of sentences handed down in courts. Many of these articles focused on achieving a “record sentence” in Australia, implying that the courts are starting to use penalties on the higher end of the scale. However, many of these articles reported on the most heinous acts of animal cruelty where a high sentence was likely warranted. So, although a “record sentence” was handed out, it was probably for one of the worst acts of cruelty seen in the State.

“*Recently convicted animal-cruelty offender, [name removed] has received a record sentence for brutally stabbing, beating and hanging a dog… The longest jail term imposed for an RSPCA NSW prosecution*”(stated by journalist, Liverpool Champion New South Wales, 2 July 2019)

Many quotes from authorities were used in this theme to articulate that animal cruelty is being taken seriously and will not be tolerated. These statements were often found in articles discussing proposed amendments to strengthen the legislation and signified to the public that they should be afraid of animal welfare law, as they will be caught and charged with cruelty. It implies that the authorities that either enforce or draft the laws are concerned with animal welfare and will ensure that the legislation is where it needs to be to adequately protect animals.

“*We take incidents such as these very seriously and anyone who engages in activities such as these will face the full brunt of the law*”(quoted from NSW police spokesperson, Magnet Eden New South Wales, 3 October 2019)

“*Minister for Agricultural Industry Development and Fisheries [name removed] said animal welfare was everybody’s responsibility and Queensland would not stand for cruelty to animals*”(quoted from by Minister, North West Star Queensland, 29 October 2019)

“*A proposed crackdown on the mistreatment of pets will debated in the ACT Assembly on Tuesday, as the government seeks to pass its Australian-first animal welfare reforms*”(stated by journalist, Canberra Times, 30 July 2019)

There were mentions again to the community in this theme, however unlike the first theme, these statements were overall more positive in nature and focused on this idea that the reason why cruelty reports are increasing is because the community are becoming more aware of signs of cruelty.

“*RSPCA Chief Inspector [name removed] said the increasing number of reports was a sign that the community was learning to identify the signs of animal abuse and neglect*”(quoted from RSPCA SA spokesperson, The Sunday Mail Adelaide, 4 August 2019)

“*We will continue to work with the community in the first instance to change negative behaviors, but, when necessary, our inspectorate’s ability to take punitive and corrective measures will now be strengthened by the additional offences*”(quoted from RSPCA ACT spokesperson, Canberra Times, 27 September 2019)

Many reported animal cruelty cases are finalized by the enforcement agency working with and educating animal owners on humane pet care [54]. Statements like those above show that the community education efforts undertaken by enforcement agencies appear to be working. It also provides some educational material to the reader that many investigations of cruelty do not pursue legal criminal action, in that Australia wide only 0.0062% of cruelty investigations result in prosecution [55]. On this note, a small number of articles discussed alternative penalties to the common fine and jail sentences.

“*Sometimes when these offenders escape jail, we might need to stop and ask whether the court might have got it right. Maybe these special cases need a rehabilitative approach, rather than a prison cell, if we have any chance of curbing the disturbing behavior and protecting animals and people in the future*”(quoted from RSPCA QLD spokesperson, The Sunday Mail (Queensland), 7 August 2019)

Alternative rehabilitative forms of penalty, often in the form of court mandated counselling and conflict resolution training, have been discussed in the literature as potentially more useful forms of punishment for animal abuse [56,57,58]. However, these penalties were rarely discussed in articles included in the dataset, the emphasis mainly being on custodial sentences. Statements like the above challenge the idea that harsher sentences are needed and show some support for the current system of discretion and variety in penalty imposition, whilst still advocating for slight improvements. Such statements provide further educational material to the reader and would challenge their thinking when it comes to penalties, perhaps breaking this perceived cycle of harsher penalties in legislation equals harsher sentences in court, which equals less cruelty.

### 3.3. Theme 3: ‘Reforms Are Not Necessary’: Action Is Too Harsh

This theme provides a counterargument to the other two themes, in that animal welfare reforms are not necessary, and the legislation should remain the same without any further action. However, it was defined at various levels; some articles articulated a mild aversion towards reforms (e.g., “The Opposition is opposed to what they described as ‘radical laws’”—stated by journalist, Canberra Times, 30 July 2019), whilst others expressed extreme disdain (e.g., “[Owners] could now potentially be framed as animal abusers for failing to do things like get their dog’s nails clipped. This is neither reasonable nor fair”—quoted from Member of Parliament, Canberra Times 30 July 2019). This theme was far less common than the other two themes, only accounting for 10% of the dataset.

The majority of this theme comprised discussions on proposed amendments being too tough and impractical for everyday animal owners, and therefore not necessary. Although similar, the below quotes relate to different proposed amendments. The first quote referring to the proposed changes to animal welfare offences under the *Animal Welfare Act 1992* (ACT) after legally recognizing animal sentience. The second quote relates to a proposal to prohibit dogs being secured on metal trays over 28 degrees Celsius (82.4 degrees Fahrenheit). This law has since passed and can be found in s 6(4) of the *Prevention of Cruelty to Animals Regulations 2019* (Vic).

“*[Opposition leader] had already made clear her opposition to the proposal when she claimed the laws could effectively turn dog lovers into criminals*”(quoted from Member of Parliament, Canberra Times, 27 September 2019)

“*We want a rule to come in to make sure animals are protected but we want to make sure it’s a practical rule to work with*”(quoted from Victorian Farmers Federation spokesperson, Wimmera Mail-Times, 9 September 2019)

There were also mentions of simply using common sense when it comes to human dealings with animals, and that as a society we should not require such paternalistic laws that overlook common sense.

A less common attribute to this theme was that animal cruelty is not as bad as it seems and often prosecution is not always necessary. Most animal cruelty cases are often negligent-type acts (commonly defined as ‘basic cruelty’), rather than the application of actual cruelty (commonly defined as ‘aggravated cruelty’) [5], in which the latter under legislation have a higher maximum penalty attached [29]. Hence, the majority of cases would not warrant the highest penalty, as they fall under this ‘basic cruelty’ category. Often these articles expressed that the offenders are remorseful and, in some cases, did not know they were doing the wrong thing.

“*[RSPCA prosecutions officer] said people often thought only ‘evil’ people were charged with animal cruelty, but the vast majority were people who’d made ‘silly choices’*”(stated by journalist, The Sunday Mail Queensland, 16 September 2019)

“*We don’t just want to go and stick people in jail. We want to change behavior,*” [RSPCA spokesperson] said. “*I don’t want people to run away when they see the RSPCA inspector van; I want them to work with us so that we can continue to reach the good outcomes that we do*”(quoted from RPSCA ACT spokesperson, Canberra Times, 28 July 2019)

As articulated here, focusing on harsh sentences or penalty increases may not be necessary to punish most animal abusers, and instead advice and assistance from enforcement agencies could work in lieu of prosecution. This sentiment is similar to the previous theme discussed on alternative penalties. However, instead of recommending use of more rehabilitative forms of penalty, these statements are highlighting the option of not using the court system at all, instead working with the owners in an educative fashion.

## 4. Discussion

This study applied a thematic analysis to news articles and identified three common ways the media portrays animal cruelty reports and penalties in Australia. These themes were: (1) current animal welfare laws are not good enough and need reform; (2) laws are improving, and animal welfare is being taken seriously; and (3) reforms are too harsh and unnecessary. The three themes were established from news reports that included a discussion of, or reference to, the penalties or sentences for animal welfare offences. The remainder of this paper will discuss the agenda-setting effects these themes could have on the ‘community expectations’ of the penalty outcomes that should arise from animal welfare law, and the consequences these ‘expectations’ have on the political and legal framework underpinning human responsibilities towards animals.

### 4.1. Agenda-Setting Effects

Animal welfare laws in Australia are essentially public interest laws, in the sense that governments will generally legislate in the public interest, whilst also balancing this with advancements in knowledge about animals and their welfare needs. As a result, the community has some ability to drive legislative change through their expectations and opinions, as regularly observed during political debates [4,5]. However, as Geysen et al. [4] noted, despite reviewing the explanatory notes and second reading speeches of these political debates, the content of these ‘expectations’ which are relied upon by policy-makers, are largely unknown. Based on the established agenda-setting effects of the news media [37], the themes established in this paper will give a likely indication of the information the public are using to build these ‘expectations’, making them potential indicators of public opinion. At this level this paper is making an assumption that will require further validation with appropriate sociological research.

Based on the emerging themes, it is likely that the public are measuring the success of animal welfare laws by the magnitude of the penalty handed down in court, regardless of the details of the case. The reports on ‘record sentencing’ were prevalent in the positive theme of “laws are improving”, whereas ‘lenient sentencing’ made up the bulk of the negative theme of “laws are not good enough”. Such findings are similar to Grugan [21] results on animal cruelty media analyses, where the author found that less severe cases were reported in a more neutral athematic way, in contrast to the more severe cases which were presented emotively with loaded language that condemned either the act, the offender or the justice system. Thus, the public are encouraged to perceive severe penalties as positive, and minor penalties as negative, which is in line with reported sociological research [34]. Hence, the combined message of the two themes is that harsher penalties (often in the form of jail time) are needed to protect animals from cruelty, regardless of the type of cruelty inflicted on the animal. This is in contradiction to academic commentary which questions the effectiveness of harsher penalties [5,32,58,59], and advocates for alternative penalties which have a more rehabilitative focus to tackle the root of the problem, not the result.

With that said, whether intentional or not, the media is likely providing an element of social deterrence through this penalty portrayal. Deterrence theory focuses on the ability of laws to deter members of society from committing illegal acts, through the belief that capture and punishment of offenders will occur [60]. Simply put, the threat of punishment is enough to deter people from committing a crime. However, deterrence is only one aspect of punishment theory that underpins the imposition of penalties in court, with the other aspects of rehabilitation, retribution, restitution, and incapacitation [61,62,63,64] perhaps being less visible to the public. When the media portray harsh sentences and maximum penalties in a positive light, it signifies to the public that harsh penalties are the norm for animal welfare offences and creates social deterrence due to fear of the harsh penalties. Therefore, whilst it is likely that the media are creating elevated penal ‘expectations’ amongst the public, they may be playing a valuable role in creating a social deterrent effect.

### 4.2. Sociological Research

Sociological research investigating public opinion towards penalties for animal welfare offences is now over a decade old in Australia, thus further research is required to update our current understandings of public opinion to draw more conclusive assessments. However, the portrayal of penalties by the media is consistent with the findings of this previous sociological research, which has determined that the public favor harsh penalties for animal welfare offences, namely prison sentences [33,34]. This implies that there could be a connection between the two, however establishing causality is difficult as is it unknown which came first, public opinion or media content [17]. Public opinion could be influencing media coverage, as journalists understand it will generate interest [65], conversely media coverage could be influencing public opinion. Establishing cause-and-effect is difficult as the evidence is grounded in “real world” data that is influenced by a range of uncontrollable and unknown factors. However, given that for the majority of the public the media is their only source of information on animal welfare law, it is highly likely, based on agenda-setting theory, that the media is having some effect on public opinion, even if the extent of this influence is unknown.

Similarly to Arluke et al. [22], we observed that the majority of media reports only discussed the most severe cases of animal cruelty, likely because they will generate greater public interest. However, given that these severe or deliberate (previously defined as ‘aggravated’) cases of animal cruelty are in fact in the minority [5], this selective reporting could skew public perception on the reality of the crime of animal cruelty based on a consideration of both prevalence and severity of acts, making it appear a more significant issue that it actually is. Discussion of penalties in these articles, creates an urgency for harsher penalties given the heinous nature of the case. In addition, this urgency for harsher penalties can also be created through referral to the established link between acts of human violence and animal cruelty [46,47,48,49,50,51,52,53]. The ‘link’ suggests that offenders who are deliberately cruel to animals are more likely to be violent to humans, as evidence suggests these offenders have the potential to develop into child abusers [24,31], spouse-beaters [24,25,26,27,28], or even murderers [29,30]. The use of this messaging through the media likely creates a fear response in the public, which further solidifies the need for harsher penalties, in order to break this cycle at the animal stage, before it progresses further to humans. However, as mentioned above, these type of offences are in the minority, with the majority of offences being the more minor duty of care breaches [5]. Thus, if the public are perceiving these severe and deliberate ‘aggravated’ cases as the norm, they could also be indirectly perceiving these harsher sentences as the norm, which in turn elevates their expectations.

### 4.3. Sentencing Guidelines

Although the public are demanding harsher sentences, as Markham [23] noted, case sentencing is a complex process, it is not as simple as sentencing based on the communities’ expectations. Case sentencing is a multifactorial process, in which judges must consider numerous factors, such as, judicial discretion, adherence to sentencing guidelines and the doctrine of precedent, when deciding a sentence. Sentencing in Australia is guided by sentencing legislation, where judges will decide a sentence in accordance with the State’s relevant Act. For example, in South Australia, Section 10 of the *Sentencing Act 2017* outlines the use of imprisonment and states that prison sentences must only be handed out as a last resort or if it is required for community safety. Considering the majority of animal cruelty cases are breach of duty of care cases, not aggravated cases, where the offender intends to harm the animal [5], it implies (at least from the perspective of violence) that the majority of animal cruelty offenders do not pose a risk to the community. In addition to adhering to sentencing guidelines outlined in legislation, judges are also bound by precedents and must follow the determinations made in higher courts [66], this makes any substantial increases to sentences difficult if precedent binds judges to sentence in accordance with previous decisions. Instead, a slow increase to sentences may occur overtime, instead of the large increase the public appear to favor.

Thus, even though the public favor prison sentences, they are not necessarily warranted for the majority of perpetrators of animal offences. Therefore, the penalty reforms occurring throughout Australia are essentially symbolic gestures [4,32], used to signify the ideology of “getting tough” on offenders [30,31], rather than practical efforts to elevate court-handed down penalties [5].

### 4.4. Penalty Reform Cycle

Through combining findings from sociological research, evidence of agenda-setting from this study, and the content of sentencing guidelines in Australia, we posit that there is a relationship between all three. This relationship forms the basis of our proposed ‘penalty reform cycle’ (Figure 1), which is hypothesized to contribute to the public’s elevated penal expectations and this idea of an ‘enforcement gap’ in animal law, in that the expectations in relation to penalties are not aligning with the realities of the justice system [54]. This cycle speculates that the community expectations of harsher penalties is driving largely ‘symbolic’ reform efforts to raise maximum penalties for animal welfare offences, which the media report through the second theme as “laws are improving”. However, factors such as judicial discretion, the need to adhere to sentencing guidelines and statute, and resource constraints such as overcrowded prisons and the cost of incarceration, lead to harsher penalties not being applied in court, which causes the media to report on the ‘lenient sentencing’ common in the first theme of “laws are not tough enough”. The public then become disillusioned with the legal system and the penalties imposed, fueling the expectations that harsher penalties are needed, which begins the cycle again.

Although the inter-relationships between parties in this cycle are clear, it is unknown where the cycle starts due to the difficulties in establishing causation [17]. Sinclair et al. [67] established that a media exposé had an influence over public views on animal welfare. In contrast, Tiplady et al. [68] found that less than 10% of people exposed to a broadcast about animal cruelty in Australia acted and contacted politicians. However, as previously stated, given that the media is a major source of information on animal welfare matters for the public, it will likely have some impact on public opinion. In order to break or redirect the cycle, public education may be a successful intervention. Agenda-setting effects are less likely to influence readers who have knowledge or experience with animal welfare law [19,20] and have greater knowledge of the justice system and the types of animal welfare cases being presented to it. Therefore, any public education campaigns should focus on how the sentencing court considers and hands down penalties for individual offenders as well as the typical distribution of offences across the aggravated and breach of duty-of-care categories.

It should be noted that an individual’s experiences and perspectives will influence the way information is interpreted for that individual, and therefore using themes in this context does not guarantee the readers will interpret such information in the speculated way [69]. The themes in this paper are intended to provide a guide to the types of messaging apparent in news articles on animal welfare law and how they could influence ‘community expectations’, and consequently the political framework surrounding animal welfare legislative reform. However, further sociological research is needed to validate the speculations in this paper by directly exploring public opinion and expectations of animal welfare law, and the contributions those expectations have on this proposed penalty reform cycle.

## 5. Conclusions

Portrayal of animal cruelty penalties by the media in Australia can be allocated into three contrasting themes of (1) current animal welfare laws are not good enough and need reform; (2) laws are improving, and animal welfare is being taken seriously; and (3) reforms are too harsh and unnecessary. The predominant theme emerging related to magnitude of penalties, regardless of the severity of the offence. Articles discussing statutory amendments to increase penalties were often present in the second theme, whilst discussion of custodial sentences handed down in court were common to the first theme. It is proposed that there is a relationship between themes one and two, and the ‘community expectations’ that have been referred to as drivers of animal law reform efforts in Australia. This relationship forms the basis of the proposed penalty reform cycle, whereby the media reports on recent statutory amendments with a theme that ‘laws are improving’ (theme two), then, due to a wide range of judicial system-related factors, harsher sentences are not applied in court, resulting in media reporting of ‘lenient sentencing’ (theme one). Therefore, the public become dismayed at the penal system, forming the ‘community expectations’, which then inform parliamentary debate and fuel future amendment efforts. Thus, the cycle continues.

## Figures and Tables

**Figure 1 animals-12-02918-f001:**
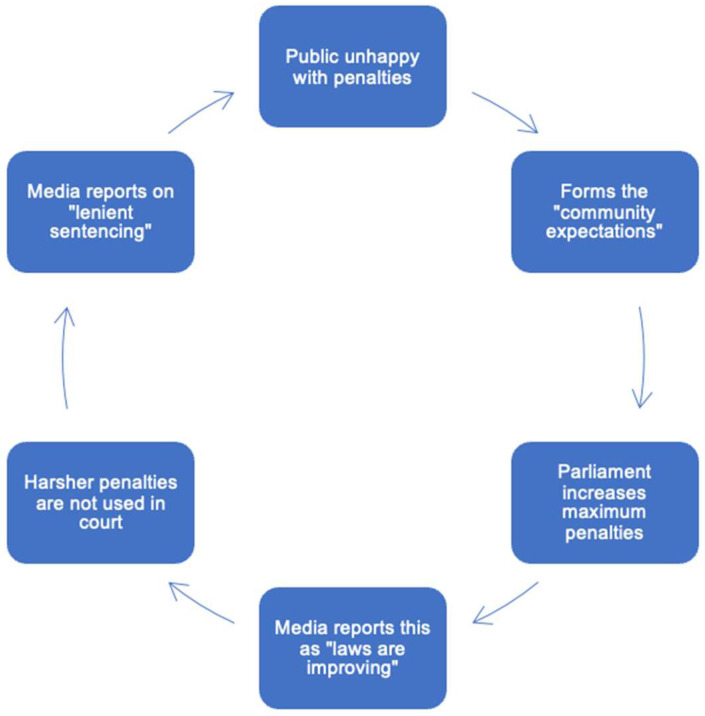
Proposed penalty reform cycle around animal cruelty offences involving the media, the public, courts, and the legislature. “Laws are improving” comprises of the second theme found from the analysis, whilst “lenient sentencing” comprises of the first theme.

## Data Availability

Not applicable.

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
