# Peer review of "Portraying Animal Cruelty: A Thematic Analysis of Australian News Media Reports on Penalties for Animal Cruelty"

_animals, 2022, doi:10.3390/ani12212918_

Round 1
Reviewer 1 Report
I believe this is a novel study that contributes to a growing body of literature on media depictions of animal cruelty and related issues of legal and jurisprudence responses to these acts. I would recommend acceptance after minor revisions, which would include:
1. Removal or revision of Table 1. It seems unnecessary to the manuscript in its current form since the authors do not refer back to or cite these specific statutes later in the paper. Citation of each statute in the reference list is also unnecessary unless information from these laws is cited elsewhere in the manuscript. I would suggest either identifying the statutes in the discussion of themes where appropriate, identifying the frequency with which each statute was addressed in the article sample and include that in this table, or simply removing the table.
2. A related issue to the above – on Page 8, Lines 380 – 384, the quotes here seem to be regarding the same proposed amendment based on discussion of dog owners and the dates of publication for each article. If this is the case, I would recommend that the author(s) identify that these quotes are, indeed, related to the same proposed law change for transparency in their analysis.
3. An issue of conceptual clarity: some of the exemplar quotes the author(s) include to illustrate the themes seem to be demonstrative statements made by whomever was the journalist writing the article, some others seem like quotes or sound bites from people who may have been interview by the journalist for the article. If this is the case, it would be important to not only identify which is which but also include some information on anyone who is quoted in an article in terms of their role as an official or member of the public. Quotes from people involved in the case or members of the public in reaction to the case may involve very different messaging from one another and from that of the journalist writing a piece on this issue.
I believe once the author(s) address these minor issues, this manuscript can be accepted for publication.
Author Response
Removal or revision of Table 1. It seems unnecessary to the manuscript in its current form since the authors do not refer back to or cite these specific statutes later in the paper. Citation of each statute in the reference list is also unnecessary unless information from these laws is cited elsewhere in the manuscript. I would suggest either identifying the statutes in the discussion of themes where appropriate, identifying the frequency with which each statute was addressed in the article sample and include that in this table, or simply removing the table.
Thank you for this suggestion, the table has now been removed.
A related issue to the above – on Page 8, Lines 380 – 384, the quotes here seem to be regarding the same proposed amendment based on discussion of dog owners and the dates of publication for each article. If this is the case, I would recommend that the author(s) identify that these quotes are, indeed, related to the same proposed law change for transparency in their analysis.
Thank you for picking this up. While the quotes related to different proposed amendments, we have added a discussion on lines 444-450 to make this clear to the readers and explain the types of amendments they were referring too.
An issue of conceptual clarity: some of the exemplar quotes the author(s) include to illustrate the themes seem to be demonstrative statements made by whomever was the journalist writing the article, some others seem like quotes or sound bites from people who may have been interview by the journalist for the article. If this is the case, it would be important to not only identify which is which but also include some information on anyone who is quoted in an article in terms of their role as an official or member of the public. Quotes from people involved in the case or members of the public in reaction to the case may involve very different messaging from one another and from that of the journalist writing a piece on this issue.
We do agree with this point and have now included this information as ‘stated by’ or ‘quoted from’ after every quote.
Reviewer 2 Report
The topic is good, but as it is, it is too superficial and raises more questions than answers.
For the sake of the international readers, the authors should tell about the punishment scales and what crimes against animals exist in Australian laws. Otherwise, it is difficult for readers to understand the topic.
What was to research question you wanted to answer? The authors wrote: "To understand how penalties are portrayed in the media." - why the authors wanted to understand that? Please clarify.
You wrote: ”Metropolitan and rural newspapers were included to reduce any demographic biases between the two news sources and identify a newspaper sample with broad readership demographics and political orientations.” Did the newspapers include papers from farmers interest organizations? They often have a different view on the matter than those who live in large cities. It would be necessary to explain the background of the selected magazines. Newspapers representing different interests can also skew the result if the papers are not chosen equally.
In addition, the distribution of the number of articles into each of the different groups should be multiplied.
A scientific article should have clear conclusions, now the conclusions section just repeats what had already been said. At the beginning of the study you should clearly tell what the research problem is, and then at the end answer to the problem.
What is the novelty value of the study? As a whole, the evaluation research remained incomplete and superficial. It needs deepening and sharpening (analytical and reflective).
Author Response
For the sake of the international readers, the authors should tell about the punishment scales and what crimes against animals exist in Australian laws. Otherwise, it is difficult for readers to understand the topic.
Thank you for this comment, we have added a summary of this information on lines 109-113 in the introduction. It is only a summary because each of the eight states/territories have different maximum penalties and offences, making it very complicated and outside of the focus of this paper to explain in detail. We have added a reference our previous work that details these maximum penalties, and offences for the readers to refer to.
What was to research question you wanted to answer? The authors wrote: "To understand how penalties are portrayed in the media." - why the authors wanted to understand that? Please clarify.
We have amended the final paragraph of the introduction to make our aim clearer (see lines 135-147). Given the media is major influencer of public opinion, we wanted to understand how penalties are portrayed to the public via this primary information source as agenda-setting literature suggests that the public will be using this information to inform their opinions. Knowledge of this is useful since previous sociological research of Australian public opinion suggests that the public are displeased with the current penalties. Given that governments have increased their maximum penalties, something else must be at play to lead to the formation of this opinion. Therefore, we are investigating media portrayals to see if there could be a link between public opinion of animal cruelty penalties and media portrayals. We hypothesise that there is a link and have presented this in our suggested penalty reform cycle (from line 613).
You wrote: ”Metropolitan and rural newspapers were included to reduce any demographic biases between the two news sources and identify a newspaper sample with broad readership demographics and political orientations.” Did the newspapers include papers from farmers interest organizations? They often have a different view on the matter than those who live in large cities. It would be necessary to explain the background of the selected magazines. Newspapers representing different interests can also skew the result if the papers are not chosen equally.
Our search was limited to only Australian print media available on the Newsbank database, and our search criteria specifically excluded any magazines/journals/newsletters. We have added this information on line 181-184. As most of these farmer interest groups produce magazines/newsletters, they were automatically excluded from the search. However, there are some quotes that were used from these groups in the newspapers we analysed. We have now added information on the source of the quote, where individual quotes are presented in the results.
In addition, the distribution of the number of articles into each of the different groups should be multiplied.
Our apologies, but we are unsure which groups you are referring too here as we have not divided our data into groups. If the reviewer could provide a little more information into this comment, we would be happy to consider it further.
A scientific article should have clear conclusions, now the conclusions section just repeats what had already been said. At the beginning of the study you should clearly tell what the research problem is, and then at the end answer to the problem.
Our research question was to understand how penalties are portrayed in the media as literature suggests that they are influencing public opinion. We have attempted to clarify this question in the introduction. We used an interpretative research paradigm employing thematic analyses to understand their portrayal and found that penalties are portrayed in three different ways (three themes). We then used our findings and previous literature to propose a relationship between these themes, community expectations and law reform (being the penalty reform cycle). Our conclusion restates our findings (three themes of portrayal) and summarised our proposal for this penalty reform cycle. We respectfully suggest that we have answered the research question we posed, and have proposed a reform cycle which generate further hypotheses for future empirical work to test.
What is the novelty value of the study? As a whole, the evaluation research remained incomplete and superficial. It needs deepening and sharpening (analytical and reflective).
We advance that this study is novel as it is the first Australian research into the portrayal of animal cruelty in the news media (where the previous studies were US-focused, which are referenced and discussed throughout our paper - Grugan (2019) and Arluke et al (2002)). Our assertion of novelty seems to be shared by the other reviewers. This study is also the first to specifically focus on penalties and how they are portrayed by the media. We have put forward a (new we believe) suggestion on how public opinion, media reporting and law reform could be linked.
Reviewer 3 Report
Overall, this is an extremely interesting article and an enormous amount of work has gone into it. Systematic reviews are a lengthy process, and this has added something very valuable to the field. The comments below are intended to make the piece stronger and I hope they are appreciated by the authors.
· It would be beneficial to the reader to have an explanation or definition of ‘community expectations’ and how they relate to animal welfare, or at least what it means to the authors.
· At p.2 there are two different spellings of analyse*, but am unsure as to whether that is intentional.
· Inescapable resource constraints – could be developed slightly further.
· It may be better to have a more distinct and detailed background section, which develops the key concepts and outlines previous studies in more detail, to better place the data in the context of the field. There are also elements of findings brought up in the analysis, which are not covered in the introduction/background, such as the link between human and animal violence. They need to be more grounded in the literature and more developed in the discussion. This is the same for punishment theory, as most of the finding relate to sentencing and punishment and it is not covered in the background.
· In materials and methods, it would be useful to know how many online national print media sources were searched. This would be helpful for international readers who do not know how many online media sources in Australia exist.
· Analysis – it would be useful to know if all authors were involved in this stage.
· The findings/analysis section is well written. The discussion could be more developed and relate more to some of the issues raised in the findings, such as the link between human and animal violence. The proposed penalty reform cycle is very interesting, and I agree that it is difficult to identify where it starts, but it could be better developed, e.g. you highlight the need for education of animal welfare in the findings, but this fails to make its way into the discussion. Perhaps the reform cycle is a good section to do this.
Author Response
It would be beneficial to the reader to have an explanation or definition of ‘community expectations’ and how they relate to animal welfare, or at least what it means to the authors.
Thank you for this suggestion, we have added further details on lines 128-132 to better explain this concept of ‘community expectations’.
At p.2 there are two different spellings of analyse*, but am unsure as to whether that is intentional.
Thank you for picking that up, we have amended this error on line 99.
Inescapable resource constraints – could be developed slightly further.
We have changed the wording of this sentence to make it clearer and added a reference to support our statement (see line 134).
It may be better to have a more distinct and detailed background section, which develops the key concepts and outlines previous studies in more detail, to better place the data in the context of the field. There are also elements of findings brought up in the analysis, which are not covered in the introduction/background, such as the link between human and animal violence. They need to be more grounded in the literature and more developed in the discussion. This is the same for punishment theory, as most of the finding relate to sentencing and punishment and it is not covered in the background.
Thank you for this suggestion, we have considered it and believe our background as it stands currently provides the necessary details to introduce the topic of penalties for animal cruelty in Australia and used relevant theories (agenda-setting) to justify our methodological approach. We are concerned that anymore information might cause the introduction to lose direction. However, we do agree with your comments that the link between human and animal violence and punishment theory need further discussion. To rectify this, in the discussion section we have add more information on the link (see lines 569-584) and on punishment theory (see lines 537-539). We hope this will suffice.
In materials and methods, it would be useful to know how many online national print media sources were searched. This would be helpful for international readers who do not know how many online media sources in Australia exist.
Thank you for this suggestion, there were a total of 533 Australian newspapers that were searched. We have included this information on lines 182-183.
Analysis – it would be useful to know if all authors were involved in this stage.
The thematic analysis used in this study was Braun and Clarke’s (2006) reflexive thematic analysis, which advocates against inter-rater reliability measures to assess coding accuracy or quality. Therefore, only one author (the first author RM) coded and analysed the data, as per the reflexive thematic analysis methods. We have now included this information on lines 220-223.
The findings/analysis section is well written. The discussion could be more developed and relate more to some of the issues raised in the findings, such as the link between human and animal violence.
As per our response above, we have added a further discussion on the link on lines 569-584.
The proposed penalty reform cycle is very interesting, and I agree that it is difficult to identify where it starts, but it could be better developed, e.g. you highlight the need for education of animal welfare in the findings, but this fails to make its way into the discussion. Perhaps the reform cycle is a good section to do this.
Thank you for this suggestion, we have included a discussion on public education on lines 645-652.
Reviewer 4 Report
This is an interesting read. The content is relevant to the field of animal law. It adopts a clear and appropriate structure. It includes original ideas in its discussion, although the methodology involving a content analysis of animal cruelty in the news media is not novel having been carried out recently in the USA (Grugan, 2019). The authors acknowledge this and discuss Grugan.
The methodology is clearly stated and appropriate to the research. The results are reproducible based on the details provided in the methods section.
I think the article could be published as it is, but I make some suggestions for improvement:
1. It would be helpful to define the term ‘community expectations’ in this context. I have not come across this concept in Animal Law but it appears to be more common in Australia. Given the international readership of the journal, it would be helpful to define the concept at the start of the article (in line 116).
2. Your ‘penalty reform cycle’ proposes that when Parliament have increased the maximum penalties for animal cruelty offences, the courts have not used these harsher sentences and consequently the media report on lenient sentencing. It would be helpful to provide more explicit examples of new legislation that increases sentencing powers followed by case examples of the courts not using these. Whilst you give some examples of media headlines that refer to “tough new laws” you don’t provide details of these laws nor cases decided under that legislation.
3. Line 424 states that animal welfare laws are public interest laws. Does this need further clarification given that the laws also protect the animals for their own sake. The law seeks to protect animals whether or not the public know or benefit.
4. Lines 22-24 of the abstract could be rephrased so as to make a more tentative link between the media and law reform. Your statement that “it is likely that the media could be influencing recent reform …” seems bold without more evidence.
Author Response
- It would be helpful to define the term ‘community expectations’ in this context. I have not come across this concept in Animal Law but it appears to be more common in Australia. Given the international readership of the journal, it would be helpful to define the concept at the start of the article (in line 116).
Thank you for this suggestion, we have added further details on lines 128-132 to better explain this concept of ‘community expectations’ in Australia.
- Your ‘penalty reform cycle’ proposes that when Parliament have increased the maximum penalties for animal cruelty offences, the courts have not used these harsher sentences and consequently the media report on lenient sentencing. It would be helpful to provide more explicit examples of new legislation that increases sentencing powers followed by case examples of the courts not using these. Whilst you give some examples of media headlines that refer to “tough new laws” you don’t provide details of these laws nor cases decided under that legislation.
We believe we have named the bottom box in Figure 1 incorrectly and have now changed it to ‘media reports this as laws are improving’ rather than the prior ‘media reports on tough new laws’. We believe our use of the word ‘laws’ may have been misinterpreted in this context, as we were referring to the maximum penalty increases for animal cruelty offences, not any specific new legislation that has been enacted. These maximum penalty increases have consistently occurred throughout the Australian jurisdictions and have been referred throughout the paper. We have also made this clear on line 621 and in the figure legend to hopefully avoid any misinterpretation. We apologise for any confusion and thank you for bringing this to our attention.
- Line 424 states that animal welfare laws are public interest laws. Does this need further clarification given that the laws also protect the animals for their own sake. The law seeks to protect animals whether or not the public know or benefit.
We have added further clarification to this point on lines 497-500 to make it clear that while governments will listen to the public when making laws, they will also balance this with the needs of animals.
- Lines 22-24 of the abstract could be rephrased so as to make a more tentative link between the media and law reform. Your statement that “it is likelythat the media could be influencing recent reform …” seems bold without more evidence.
Thank you for picking this up. We have changed this phrasing to “the media might indirectly be influencing recent reform efforts to amend maximum penalties in Australia, through guiding and shaping public opinion.”
Round 2
Reviewer 2 Report
The manuscript has improved and the new version is acceptable.